# Lipid Droplets in Lung Cancers Are Crucial for the Cell Growth and Starvation Survival

**DOI:** 10.3390/ijms232012533

**Published:** 2022-10-19

**Authors:** Jrhau Lung, Ming-Szu Hung, Ting-Yao Wang, Kuan-Liang Chen, Chi-Wen Luo, Yuan-Yuan Jiang, Shin-Yi Wu, Li-Wen Lee, Paul-Yann Lin, Fen-Fen Chen, Hui-Fen Liao, Yu-Ching Lin

**Affiliations:** 1Department of Medical Research and Development, Chang Gung Memorial Hospital, Chiayi Branch, Chiayi 61363, Taiwan; 2Department of Pulmonary and Critical Care Medicine, Chang Gung Memorial Hospital, Chiayi Branch, Chiayi 61363, Taiwan; 3Division of Hematology and Oncology, Department of Internal Medicine, Chang Gung Memorial Hospital, Chiayi Branch, Chiayi 61363, Taiwan; 4Department of Endodontics, ChiMei Medical Center, Tainan 71004, Taiwan; 5Division of Breast Oncology and Surgery, Department of Surgery, Kaohsiung Medical University Chung-Ho Memorial Hospital, Kaohsiung 80756, Taiwan; 6Department of Diagnostic Radiology, Chang Gung Memorial Hospital, Chiayi Branch, Chiayi 61363, Taiwan; 7Department of Anatomic Pathology, Dalin Tzu Chi Hospital, Buddhist Tzu Chi Medical Foundation, Chiayi 62247, Taiwan; 8Department of Pathology, New Taipei City Tucheng Hospital, New Taipei City 23652, Taiwan; 9Department of Biochemical Science and Technology, National Chiayi University, Chiayi 60004, Taiwan

**Keywords:** lipid droplet, lipid metabolism, tumorigenesis, starvation, survival, lung cancer

## Abstract

For rapid and unlimited cell growth and proliferation, cancer cells require large quantities of nutrients. Many metabolic pathways and nutrient uptake systems are frequently reprogrammed and upregulated to meet the demand from cancer cells, including the demand for lipids. The lipids for most adult normal cells are mainly acquired from the circulatory system. Whether different cancer cells adopt identical mechanisms to ensure sufficient lipid supply, and whether the lipid demand and supply meet each other, remains unclear, and was investigated in lung cancer cells. Results showed that, despite frequent upregulation in de novo lipogenesis and the lipid transporter system, different lung cancer cells adopt different proteins to acquire sufficient lipids, and the lipid supply frequently exceeds the demand, as significant amounts of lipids stored in the lipid droplets could be found within lung cancer cells. Lipid droplet surface protein, PLIN3, was found frequently overexpressed since the early stage in lung cancer tissues. Although the expression is not significantly associated with a specific gender, age, histology type, disease stage, and smoking habit, the frequently elevated expression of PLIN3 protein indicates the importance of lipid droplets for lung cancer. These lipid droplets are not only for nutrient storage, but are also crucial for tumor growth and proliferation, as well as survival in starvation. These results suggest that manipulation of lipid droplet formation or TG storage in lung cancer cells could potentially decrease the progression of lung cancer. Further exploration of lipid biology in lung cancer could help design novel treatment strategies.

## 1. Introduction

Lipids are a group of polar solvent-insoluble molecules which include fatty acids, phospholipids, sterols, and wax. These molecules, presenting either hydrophobic or amphiphilic properties, are generated directly from either ketoacyl or isoprene units [1]. Since lipids perform many indispensable cellular functions, such as structural barriers, energy storage, and signaling transfer, their uptake, biosynthesis, and utilization are tightly controlled, and the deregulations of these processes are associated with many different diseases, including cancers [2]. 

Fatty acids are precursors for many important lipids, especially phospholipids. Most adult human cells do not synthesize fatty acids themselves but acquire these molecules from the circulatory system. Only certain adult cells, including stem cells, adipocytes, hepatocytes, and cells in the lactating mammary gland, could perform de novo fatty acid biosynthesis [3,4]. Exceptions can be found in cancers, which reprogram their metabolic pathways and turn on the synthesis of fatty acids to ensure a steady supply of lipids [5,6]. The immediate upstream precursor used for fatty acid biosynthesis, acetyl-CoA, can be produced from citrate by ATP citrate lyase (ACLY), or from acetate by cytoplasmic acetyl-CoA synthetase (AceCS1). The acetyl-CoA molecule is subsequently converted into malonyl-CoA through carboxylation by acetyl-CoA carboxylases (ACC). By sequential condensation of seven units of malonyl-CoA with a priming acetyl-CoA via fatty acid synthase (FASN), palmitate is generated. The 16-carbon saturated fatty acid can be further elongated or desaturated to form different long-chain fatty acids in the endoplasmic reticulum in multiple enzymatic steps. These fatty acids can be further converted into different lipids, including phospholipids, for various cellular functions or processes [7]. Many signals regulating fatty acid biosynthesis are through the transcription via sterol regulatory element-binding proteins (SREBPs) [8]. Despite the pivotal role of fatty acids in different cellular processes, an excess of free fatty acids is harmful to cells [9] and will be rapidly converted into a neural lipid, triacylglycerol (TG), via sequential catalysis of glycerol phosphate acyltransferase (GPAT), acylglycerol-phosphate acyltransferase (AGPAT), phosphatidic acid phosphatase (Lipin), and diacylglycerol acyltransferase (DGAT) through the glycerol 3-phosphate pathway [10]. 

Storage of neutral lipids could make them become readily available when the supply is short. A lipid droplet (LD) is a specialized organelle that stores lipids. LD comprises a neutral lipid core and is wrapped with a monolayer of phospholipids and many specialized surface proteins, such as perilipins [11]. It can be found ubiquitously in all cells [12], though generally in small quantities. To make lipids readily available for cells, a close association between LDs and other organelles, including mitochondria, ER, peroxisome, endosome, and nucleus, can be frequently observed [13]. LDs are not only a temporary site for storage of carbon source, energy, and reducing power, but also an important hub for recycling lipids and proteins from organelles, and are involved in many cellular processes, such as ER stress, and autophagy [13]. LDs have been found in several types of cancers, including breast, colon, prostate, and renal cell carcinoma. They are also associated with the aggressiveness of phenotype and drug resistance [14] via numerous mechanisms, including: increasing the production of pro-tumorigenic lipid messengers [15]; absorbing hydrophilic xenobiotics, including many anticancer drugs, [16,17]; reducing ROS production by sequestering fatty acids [18]; and supporting energy production [14]. Although the upregulation of lipogenesis is widely observed in lung cancers [19], the presence and function of LDs in lung cancer remains elusive. 

Although the reprogramming of lipid metabolism and the lipid uptake system to meet the demand from rapid cell growth and proliferation in lung cancer is well investigated, whether different lung cancer cells adopt identical mechanisms to ensure sufficient lipid supply, and whether the lipid demand and supply match each other, remains unclear. This study investigated lipid biosynthesis, uptake, and storage in lung cancer cells. Results showed that different lung cancer cells adopt different proteins to acquire sufficient lipids, and the lipid supply frequently exceeds the demand, as significant amounts of lipids stored in the lipid droplets could be found within lung cancer cells. Pharmacological depletion of LDs and inhibition of lipid utilization have great impacts on cell growth and proliferation, and on survival in starvation of lung cancer cells, respectively. Further investigation of the roles of lipids in lung cancer may promote finding a way to help decelerate lung cancer progression metabolically.

## 2. Results

### 2.1. Expression of Molecules Involved in Lipid Biosynthesis and Uptake in Lung Cancer Cells

To investigate the expression of molecules involved in the lipid supply system, including lipid biosynthesis, and uptake in lung cancer cells, the protein levels of related molecules were measured using immunoblot analysis (Figure 1). Many enzymes in the fatty acid biosynthesis were found upregulated in many lung cancer cells, including the rate-limiting enzymes, ATP citrate lyase (ACLY), acetyl-CoA carboxylase (ACC), fatty acid-synthase (FASN), and phosphatidic acid phosphatase (Lipin1). Some of their activities were even augmented post-translationally by phosphorylation, such as the Ser455 phosphorylation of ACLY, which eradicated the allosteric inhibition by citrate. These results indicated that upregulation of fatty acid biosynthesis could be a common phenomenon in lung cancer. The upregulation of molecules involved in fatty acid and lipid uptake is also frequently observed in these lung cancer cells. As shown in Figure 1B, fatty acid translocase (CD36) is overexpressed in many examined lung cancer cells, including PC9, HCC827, and H441 cells. The cytosolic fatty acid binding proteins (FABPs), which could help retain fatty acids within cells, are upregulated in H1975, H1299, and H460 cells. Moreover, the expression of low-density lipoprotein receptor-related proteins 5 and 6 (LRP5 and LRP6) are also upregulated in several lung cancer cells compared with the normal lung fibroblast cell, WI-38. Rapid lipid uptake can be observed using fluorescence-labeled low-density lipoprotein (Figure 1C), as the low-density lipoprotein receptor (LDLR) involved triacylglyerol (TG) and cholesterol uptake [20,21]. The frequently simultaneous upregulation of de novo lipogenesis and lipid uptake suggested that a sufficient supply of lipids is important for lung cancer cells, but different lung cancer cells adopt different proteins to acquire a sufficient lipid supply.

### 2.2. Expression of Transcriptional Regulators Related to Lipid Biosynthesis

Since the lipid biosynthesis pathways are reprogramed and upregulated in these lung cancer cells, the expressions of several key transcriptional regulators, including CCAAT/enhancer-binding proteins (C/EBPs), hypoxia-inducing factor (HIF-1α), sterol regulatory element-binding protein/factor-1 (SREBP-1), peroxisome proliferator-activated receptor (PPARs) [22], and regulating lipogenesis, were investigated. As seen in Figure 2, different cancer cells tend to adopt different transcriptional networks to control lipogenesis. The expressions of C/EBPα and PPARγ were upregulated in several cell lines, and SREPB1, HIF-1α, PPARα, and PPARβ were more widely overexpressed in lung cancer cells.

### 2.3. Lipid Droplets in Lung Cancer Cells

In view of the upregulation of molecules involved in de novo lipogenesis and lipid uptake in lung cancer cells, the net balance between the strong lipid production/uptake and heavy utilization in lung cancer cells was investigated. LDs in the nine lung cancer cell lines were examined and compared with the normal lung cell line, WI-38, using the neutral lipid-specific dye, ORO. Since ORO can emit a weak fluorescent light signal that peaks at 613 nm when excited with 543 nm of light [23], neutral lipid storage stained by ORO in lung cancer cells was recorded under light and under a fluorescence microscope to demonstrate the specificity. As shown in Figure 3A, numerous LDs can be observed in most of the lung cancer cell lines, including HCC827, H1975, H441, H460, H1299, and A549, but with great variations in sizes and numbers. The relative amounts of neutral lipid storage in each cell line were quantitated photographically using ImageJ and are shown in Figure 3B. The presence and the number of LDs are not always associated with the harboring of driver mutation, e.g., the EGFR-activating mutation, as the LD is not present in PC9 and H1650 cells, both of which carry the EGFR exon 19 deletion mutation. A few small LDs can still be seen occasionally in those LD-deficient cell lines, including the normal lung fibroblast, WI-38, and the lung mucoepidermoid carcinoma cells, H292 cells. Since triacylglycerol is hydrophobic, its storage within cells is enveloped by a monolayer of phospholipids and some specialized structural proteins, especially members of perilipin. To investigate whether the expressions of perilipins are upregulated in lung cancer cells, the mRNA and protein expressions of these perilipins were measured (Figure 3C,D). Although the transcripts of PLIN2 and PLIN3 are detectable in these examined cells, only the PLIN3 protein is expressed in all the tested cell lines. The expression of PLIN2 protein does not correlate well with the mRNA level, suggesting that translational or post-translational regulation may be involved in the PLIN2 expression in lung cancer cells. The expression levels of PLIN2 and PLIN3 tend to be higher in lung cancer cell lines than normal ones, and the levels of PLIN3 better recapitulate the amount of LD than the levels of PLIN2, except for the high PLIN3 expression in a low LD cell line, WI-38 cells. The PLIN3 protein overexpression was also frequently found in 195 examined specimens since the early stage, but the overexpression did not differ among different gender, histology types, stages, and smoking statuses.

### 2.4. Lipid Droplet Formation Induced in Lung Cancer Cells under Lipoprotein-Deficient Growth Medium 

Despite upregulated lipid biosynthesis and uptake in lung cancer cells, LDs are absent in some cell lines, such as in PC9 and H1650 cells. To test whether LDs can be observed, cells were transferred to the lipid-deprived medium, which could induce fatty acid biosynthesis by triggering the maturation of the master regulator of lipogenesis, SREPB1 [24], and autophagy [25]. The treatment significantly altered the rate of cell growth and proliferation. The cell morphology of many cells was also changed significantly. LDs appeared in these LD-deficient cell lines, including PC9, H1650, H292, and WI-38, despite the size being relatively tiny in WI-38 (Figure 4A). The amounts of LDs were also significantly induced in other cells by the lipoprotein-deficient medium (Figure 4B). These results suggested that LD formation is an intrinsic property, and that the regulation of lipogenesis and LD formation are much tighter in normal cells than in cancer cells.

### 2.5. Blocking of Lipid Drops’ Formation Slows down the Cell Proliferation of the Lung Cancer Cells

To investigate the roles of LD in lung cancer cell growth and proliferation, LD formation was blocked pharmacologically by targeting the diacylglycerol acyl transferase (DGAT) in these cells. Previous works have shown that human cells express two DGAT isoforms, and they can compensate each other when either one of them is inhibited [26]. To completely remove LDs in cells, the DGAT1 and DGAT2 inhibitors A922500 and PF-06424439 were employed to treat lung cancer cell lines simultaneously. Complete removal or significant reduction of LDs was confirmed using the ORO staining after treatment. As shown in Figure 5, cell growth and proliferation are significantly slowed down in all the examined lung cancer cell lines but are more profound in some fast-growing cells with higher LD content in normal growth conditions, such as A549 and H460. The treatment does no harm to these lung cancer cells. These results suggest that triacylglycerol and LDs could be important for the growth and proliferation of lung cancer cells.

### 2.6. Lipid Droplets Increase Survival of Lung Cancer Cells under Starvation

Lipids constitute an important nutrient and energy reserve in cells. To investigate the cellular function of LDs in lung cancer cells, the impacts of starvation on four lung cancer cells were evaluated. As shown in Figure 6, the sensitivity to starvation correlated to the LD content in cells. The LD-deficient cell line, PC9, was much more sensitive to starvation compared with LD-containing cell lines, such as H1975, H1299, and A549. To examine whether the sensitivity to starvation was related to LDs and lipid utilization, these cells were treated with Etomoxir, an irreversible inhibitor of carnitine palmitoyltransferase-1, which blocks fatty acid utilization by inhibiting the translocation of fatty acid from the cytosol to mitochondria for energy production. As shown in the left half of Figure 6, the sensitivities of the examined cells to Etomoxir were quite mild under a normal growth medium (the second and third columns of Figure 6), but massive cell deaths were induced by Etomoxir treatment under starvation, especially in the low LD-containing cell, PC9 (the fifth and sixth columns of Figure 6). These results suggested that lipids in LD are important for the survival of lung cancer cell under starvation. 

## 3. Discussion

Lipids are essential molecules for various cellular functions, serving as physical barriers, signaling molecules, and energy storage; both sufficient supply and efficient transformation of lipids are essential for cell growth and proliferation. Owing to their pivotal roles for living cells, either deficiency or oversupply of lipids could cause diseases. Compared with normal cells, which rely exclusively on the circulatory system for lipid supply, cancer cells with huge lipid demand for much faster cell growth and proliferation may not easily be supported by the circulatory system. As such, the upregulation of de novo lipogenesis in cancers could be expected. To investigate the reprogramming of lipid metabolism in lung cancer, the expressions of proteins involved in lipogenesis and uptakes were examined in nine lung cancer cell lines and compared to the normal lung cell line, WI-38, in the current study. Results showed that not only these enzymes involved in de novo lipogenesis, but also multiple proteins for lipid uptake, such as CD36, FABP4, LRP5, and LRP6, were frequently found to be upregulated (Figure 2). Such simultaneous augmentation suggests that the lipid demand for cell growth and proliferation of lung cancer is huge, and probably neither of them alone can meet the need, as fatty acid synthase inhibitor treatment is generally harmful to cancer cells [27,28]. Moreover, a significant slowdown of cell growth and proliferation even causes deterioration in cell morphology when these cells were cultured in the lipoprotein-deficient medium (Figure 4). Although the exact extent of the increase of lipid supply from the two lipid-supporting routes was not determined, the simultaneous overexpression of multiple master regulators for lipid metabolism and uptake, including SREBP1, HIF1α, and PPARs (Figure 2), suggested that the extent of the increase could be enormous, and that a sufficient supply of lipid was very important for the growth and proliferation of lung cancer cells.

In addition to a sufficient supply of lipids, they should also be readily available. Lipid reserves in a healthy adult are mostly stored in adipose tissue distal to places utilizing them. Stores of lipids within the LDs in the cytoplasm could be found in several types of cancers and are important to their tumorigenesis. However, the expressions and functions of LDs in lung cancer remain unclear. The current study found that, despite there being a huge lipid demand in lung cancer cells, a significant number of LDs are widely present in lung cancer cells, though with some variations. The mechanism accounting for the difference in the number of LDs in lung cancer cells is not known at this moment, but the types of driver mutations seem to play a role, as the number of LDs was comparatively higher in all tested KRAS-mutated cells than in EGFR-mutated cells (Figure 3). It was found that the prognosis and survival rate are generally worse in patients with KRAS mutation than those with EGFR mutation [29,30]. Whether and how LDs promote lung cancer progression, and whether LDs can become a biomarker to differentiate different driver mutations, merits further investigation. Despite some differences, the widespread presence of LDs might suggest that a readily available lipid reserve is critical for lung cancer cells. In agreement with this speculation, the pharmacological depletion of LDs by DGAT1 and DGAT2 inhibitor treatment significantly slows down cell growth and proliferation of lung cancer cells, especially those with higher LD contents, such as A549 and H460 (Figure 5). Although the depletion of LDs is not harmful to these cells under normal condition, this treatment might contribute to a slowdown of lung cancer progression. While a two- to three- day simultaneous treatment of DGAT1 and DGAT2 inhibitors administered twice daily to mice immediately after a 4-week high-fat diet could significantly decrease intestinal TG secretion into the blood circulatory system [31], severe watery diarrhea and sporadic death after this treatment could hinder the pursuit of slowing down lung cancer progression by DGAT inhibition. Adjustment of the regimen to simultaneous administration of the two inhibitors to mice once daily (3 mg/kg PF-04620110 and 7.5 mg/kg PF-06424439) with an ad libitum normal diet for 7 days could significantly reduce the plasma TG level 4 h after drug administration. However, the plasma TG reduction effect could not be seen after tumor implantation in mice with LLC1 cells, nor were the tumor sizes consistently decreased. Further optimization of the regimen of treatment is still needed to confirm whether pharmacological elimination of LDs could be adopted to slow down lung cancer progression.

Although the depletion of LDs does not preferentially kill lung cancer cells in normal growth conditions, it may become critical when the supply of lipids suddenly turns into a shortage. As shown in the starvation treatment (Figure 6), cell lines with more LDs, including H1975, H1299, and A549, are more resistant to starvation than cell lines with less LDs, such as H1650. This result indicated that lipids are an important nutrient source for lung cancer cells under starvation. Moreover, the blocking of fatty acid oxidation by irreversible carnitine palmitoyltransferase-1 inhibitor, etomoxir, induces massive cell death regardless of the number of LDs before starvation. Although the present findings revealed the importance of lipids for lung cancer cell growth, proliferation, and survival, the roles of lipids in lung cancer remain controversial, as obesity was found to be a good prognosis marker for lung cancer patients [32,33]. Since adipose tissue may exert multiple effects on both tumor and host simultaneously, patient survival may not be decided by the metabolic effect of lipids alone. Further investigation is still needed to examine whether manipulation of lipids or LDs could slow down lung cancer progression.

Some limitations of the study should be noted to further improve the study. First, the study used a lung cell line of non-epithelial origin, WI-38, as a reference to measure the expressions of the lipid-related proteins and compare them to lung cancer cell lines of epithelial origin. Although the noncancerous normal fibroblasts adjacent to the tumor tissue are frequently adopted as normal parts in the pathology and immunohistology interpretation, it would be better to adopt normal lung cell lines of epithelial origin for future study. Second, the study adopted a fluorescence-labeled low-density lipoprotein instead of a very low-density lipoprotein to monitor the TG uptake in cancer cells. It is believed that LDL is responsible for cholesterol delivery, but apolipoprotein E (ApoE) receptors, including VLDLR, LDLR, and LRP, are all involved in the clearance of triglyceride in plasma [20,21]. VLDLR deficiency in mice minimally affects plasma lipoproteins and results only in a delay in the clearance of postprandial TG. Although monitoring LDLR activity could represent the TG uptake, using fluorescence-labeled VLDL would better demonstrate the real TG uptake. Third, the study used DGAT1/2 inhibitors instead of RNA interference to eliminate LDs. Theoretically, the elimination of lipid droplets could be achieved by either blocking the synthesis of the coat proteins or the storage content of lipid droplets. Considering that PLIN2 and PLIN3 are ubiquitously expressed in different tissues and cells, knocking down either one of them could only reduce the numbers of the lipid droplets, but not eliminate them [34,35]. Despite the convenience of the potent and specific DGAT1/2 inhibitors easily removing lipid drops, the eradication of lipid droplets by knocking down lipid droplets’ coat proteins, PLIN2 and PLIN3, simultaneously in a parallel experiment, could further increase the strength of evidence of the study.

In summary, this study investigated the lipid production, uptake, and storage in lung cancer cells and found that both lipid production and uptake systems were frequently upregulated, presumably to ensure sufficient supply for rapid growth and proliferation. Different lung cancer cells tend to adopt different proteins to acquire sufficient lipids. Despite there being a larger lipid demand in lung cancer cells than in normal cells, a significant amount of readily available lipids deposited in LDs could be found widely present in lung cancer cells. Pharmacological depletion of LDs in normal condition and inhibition of lipid utilization under starvation could slow down lung cancer cell growth and proliferation and induce massive cell death, respectively. These results indicated that lipids and LDs are important for lung cancer cell growth and survival. Further investigation of lipid metabolism and utilization in lung cancer could help develop strategies for slowing down lung cancer progression metabolically.

## 4. Materials and Methods

### 4.1. Cell Culture

A549, H292, H441, H460, H1299, H1650, H1975, HCC827, and PC9 cells were grown in RPMI 1640 medium, supplemented with 10% fetal bovine serum, 100 mg/mL streptomycin, and 100 U/mL penicillin. WI-38 cell was grown in MEM medium supplemented with 1% nonessential amino acids, 10% FBS, streptomycin, and penicillin. Cells were maintained at 37 °C and in 5% CO_2_.

### 4.2. Oil Red O Staining and Quantitation

The Oil Red O (ORO) stock solution was freshly prepared by dissolving 300 mg ORO powder into 100 mL isopropanol. The ORO working solution was then made by diluting ORO stock solution by 60% with deionized water. Immediately before staining, cells were fixed with 10% formalin and incubated with 60% isopropanol in deionized water for 5 min. Then, the ORO working solution was applied to cells for 1 min and washed in deionized water. Cell images were taken by ZEISS Axio observer fluorescence microscope (Zeiss, Oberkochen, Germany) under both bright field and rhodamine channels. The amount of LDs was quantitated by measuring the intensity of red fluorescence using ImageJ [36]. 

### 4.3. Annexin V/propidium Iodide (PI) Staining Assay

Cells (1.8 × 10^5^) were seeded to the wells of a six-well plate and cultured overnight, followed by replacement of medium containing various doses (100 and 200 μM) of etomoxir and incubation for 48 h. Cells were harvested by trypsin-EDTA detachment and washed with ice-cold PBS. Cell pellets were then resuspended in the binding buffer, stained with Annexin V FITC assay kit (#600300, Cayman Chemical, Ann Arbor, MI, USA), and analyzed by the FACS Canto II (BD, Franklin Lakes, NJ, USA).

### 4.4. Immunoblot Analysis

Cells (1.8 × 10^5^) were seeded to the wells of a six-well plate and cultured overnight. The growth medium was replaced with a new one the next morning, and the cell lysates were harvested with RIPA buffer 72 h later. Protein samples were resolved by SDS-PAGE electrophoresis and transferred onto the PVDF membrane. To prevent non-specific binding of detection antibody, the PVDF membrane was blocked by the blocking buffer at room temperature for 1 h with shaking before detection. The appropriate number of primary antibodies diluted in the blocking buffer were then added to incubate the membrane at 4 °C overnight. The excess and non-specific binding of primary antibodies to the PVDF membrane were then removed by three PBST washes. For signal detection, the membrane was incubated with the HRP-conjugated secondary in the blocking buffer at room temperature for 1 h. After removal of excess and unbound secondary antibodies by three PBST washes, the protein signal was developed by incubating with the LumiFlash Ultima chemiluminescence substrate (Visual Protein, Taipei, Taiwan) and recorded by G:Box XX9 (Syngene, Cambridge, UK).

### 4.5. Chemicals and Antibodies

The sources of chemicals used in this study were as follows: Etomoxir (#11969, Cayman). Oil Red O (#12989, Alfa Aesar, Ward Hill, MA, USA). The sources of antibodies and folds of antibody dilutions used in this studies were as follows: acetyl-CoA carboxylase (ACC) (#3676; Cell Signaling Technology, Danvers, MA, USA; 1:1000), cytoplasmic acetyl-CoA synthetase (AceCS1) (#3658; Cell Signaling Technology; 1:1000), long-chain acyl-coenzyme A (CoA) synthetase (ACSL1) (#9189; Cell Signaling Technology; 1:1000), phospho-ATP citrate lyase (#4331; Cell Signaling Technology; 1:1000), ATP citrate lyase (#4332; Cell Signaling Technology; 1:1000), CCAAT/enhancer binding protein α (C/EBPα) (#8178; Cell Signaling Technology; 1:1000), SREBP-1 (#sc-398394; Santa Cruz, Dallas, TX, USA; 1:1000), fatty acid binding protein 4 (FABP4) (#3544; Cell Signaling Technology; 1:1000), fatty acid synthase (FASN) (#3180; Cell Signaling Technology; 1:1000), peroxisome proliferator-activated receptor alpha (PPAR α) (#sc-398394; Santa Cruz; 1:1000), peroxisome proliferator-activated receptor beta (PPAR β) (#sc-74517; Santa Cruz; 1:1000), peroxisome proliferator-activated receptor gamma (PPAR γ) (#2435; Cell Signaling Technology; 1:1000), lipin 1 (#14906; Cell Signaling Technology; 1:1000), LRP6 (#3395; Cell Signaling Technology; 1:1000), CD36 (#14347; Cell Signaling; 1:1000) LDL receptor related protein 5 (LRP5) (#sc-390267; Santa Cruz; 1:1000), low density lipoprotein receptor (LDLR) (#sc-18823; Santa Cruz; 1:1000), hypoxia-inducible factor 1-alpha (HIF-1α) (#sc-13515; Santa Cruz; 1:1000), diacylglycerol:acyltransferase 1(DGAT1) (#sc-271934; Santa Cruz; 1:1000), diacylglycerol:acyltransferase 2 (DGAT2) (#sc-293211; Santa Cruz; 1:1000),PLIN2 (GP40; Progen, Heidelberg, Germany; 1:1000), PLIN3 (GP30; Progen, Heidelberg, Germany; 1:1000), and β-actin (ACTB)(#A5441; Sigma-Aldrich, Saint Louis, MO, USA; 1:10,000).

### 4.6. Immunohistochemistry

For PLIN3 protein detection in the patient samples, the anti-Perilipin 3 (N-terminus) guinea pig polyclonal antibody (Progene,) was used. Staining was performed on tissue microarray carrying 195 lung tumor specimens collected between 2009 and 2017, with signed informed consent after received the approval from the Institutional Review Board (approval no. 201602010B0). The staining procedure was started with deparaffinization using xylene and graded ethanol solution, antigen retrieved in the boiling buffer for 20 min, and brief hydrogen peroxide treatment to inactivate endogenous peroxidase. After rinsing the slides with PBS, the PLIN3 antibody solution containing 1/100 primary antibody in PBS buffer with 1% BSA and 0.5% triton X-100 was applied on the slides, covered with paraffin sheets, and incubated at 4 °C overnight within a humidified chamber. The non-specific binding of antibody was removed using 1× PBS buffer with 0.5% triton X-100, the PLIN3 expression was visualized using 10,000× dilution of peroxidase-conjugated AffiniPure Donkey Anti-Guinea Pig IgG (H+L) (Jackson ImmunoResearch, West Grove, PA, USA) with DAB. The results were interpreted by pathologists blinded to the clinical information of the studied patients, and the intensity was classified in a four-tier system: 0, no staining; 1+, weak; 2+, moderate; and 3+, strong.

## Figures and Tables

**Figure 1 ijms-23-12533-f001:**
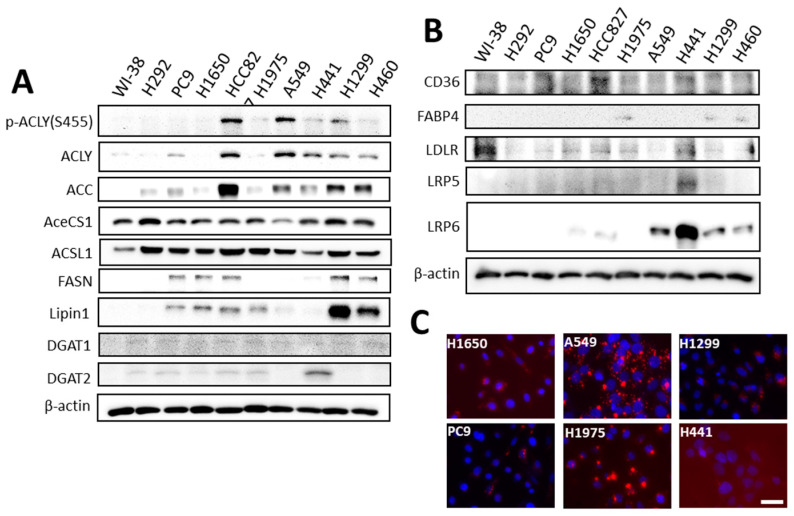
Expressions of proteins involved in lipogenesis and lipid uptake in lung cancer cells. The relative abundance of target proteins was measured using nine lung cancer cell lines and compared with the normal lung cell line, WI-38. The total protein was harvested when cell growth reached 80% complacence. The expressions of genes involved in lipogenesis (**A**) and lipid uptake (**B**) were measured using immunoblot analyses. The lipid uptakes in the lung cancer cells were recorded 2 h after incubation with the fluorescence-labeled low-density lipoprotein (BODIPY™ FL labeled LDL, Invitrogen, Waltham, MA, USA) and labeled in red. Cell nuclei were counterstained with Hoechst 33342 and labeled in blue. Images were taken at 400× magnification (scale bar, 50 µm) (**C**).

**Figure 2 ijms-23-12533-f002:**
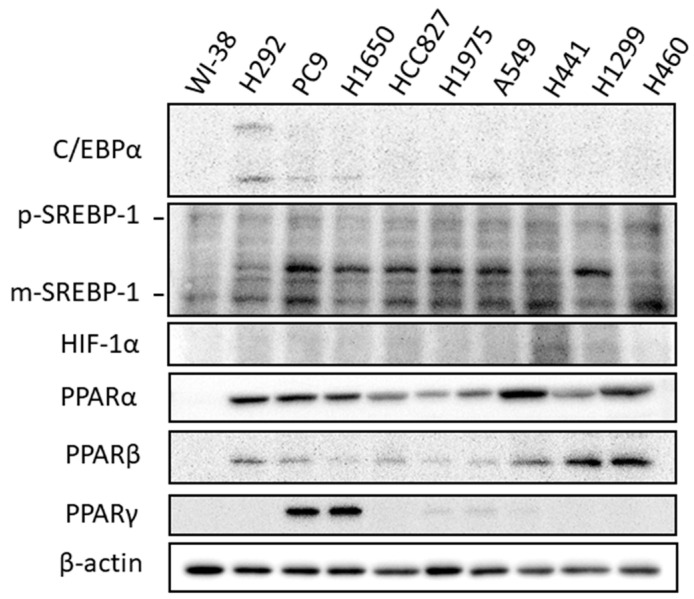
The expressions of transcriptional regulators involved in regulation of lipogenesis and lipid storage in lung cancer cells. The expressions of these transcriptional factors were compared between nine lung cancer cell lines and the normal lung cell line, WI-38, and measured using immunoblot analyses when cell growth reached about 80% confluence. The input cell lysates were adjusted into an approximately equal β-actin signal intensity and were used as a basis to compare target protein expressions on the same blot.

**Figure 3 ijms-23-12533-f003:**
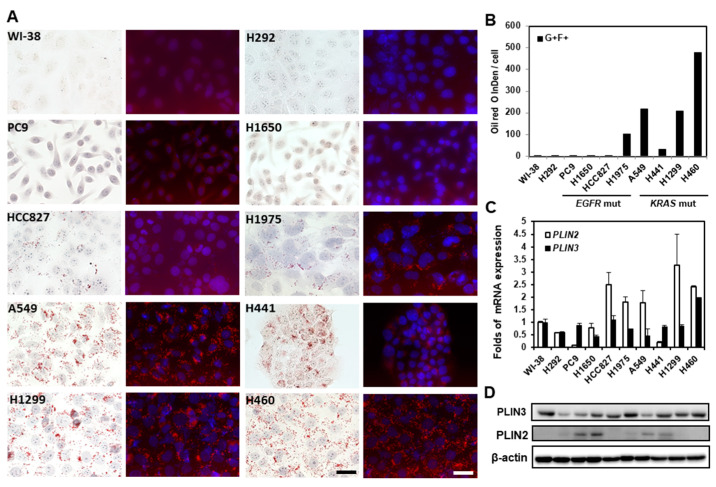
Oil droplet staining in lung cancer cell lines. (**A**) LDs in lung cancer cells were stained by ORO and recorded using Zeiss Axio Observer inverted light fluorescence microscope with Axiocam 512 color and 512 mono digital cameras. The fluorescence color emitted by Oil Red O was captured using a rhodamine filter set (Ex: 548 nm/Em: 580 nm, labeled in red) with nuceli counterstained with DAPI (labeled in blue). Images were recorded at 400× magnification (scale bar, 50 µm) (**B**) The amounts of LDs were measured using ImageJ. The fluorescence intensities of Oil Red O were integrated (IntDen value). The mRNA (**C**) and protein (**D**) expressions of PLIN2 and PLIN3 in the 10 cells were measured using real-time quantitative PCR and immunoblot analyses, respectively. The fold of mRNA expression change was calculated using 2^−ΔΔCt^ method with β-actin as the reference gene and compared with the expression level in WI-38 cells.

**Figure 4 ijms-23-12533-f004:**
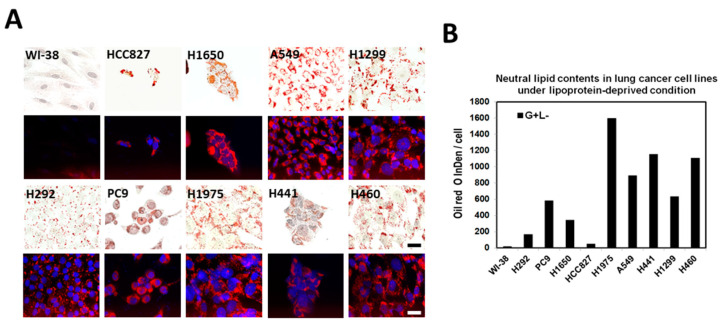
Lipid droplet formation induced by lipoprotein-deprived medium. Lung cancer cell lines are seeded into a lipoprotein-depleted medium. After seven days of growth, cells were fixed and stained with ORO and DAPI to stain LDs (labeled in red) and nuclei (labeled in blue), respectively. Images were captured at 400× magnification (scale bar, 50 µm) (**A**). The neutral lipid contents were quantitated using ImageJ as described previously (**B**).

**Figure 5 ijms-23-12533-f005:**
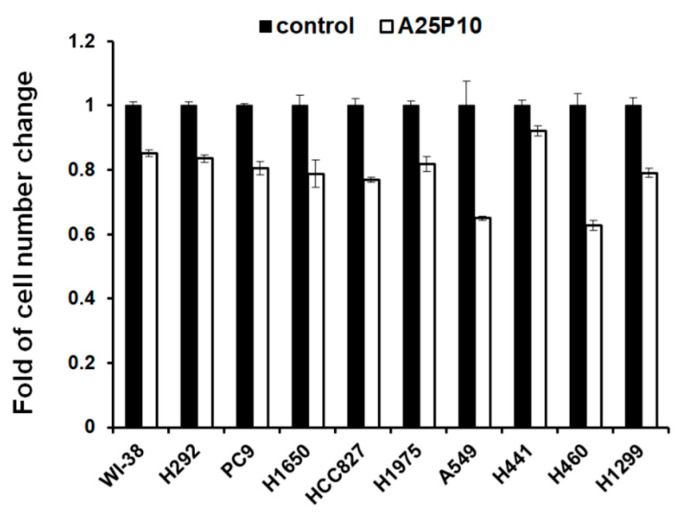
DGAT inhibitors slow down cell growth proliferation. 6000 cells were seeded into the wells of a 96-well plate and treated with DGAT1 and DGAT2 inhibitors A922500 (25 μM) and PF-06424439 (10 μM) for 72 h. Numbers of cells were measured using methylene blue staining and quantitated at absorbance at 650 nm.

**Figure 6 ijms-23-12533-f006:**
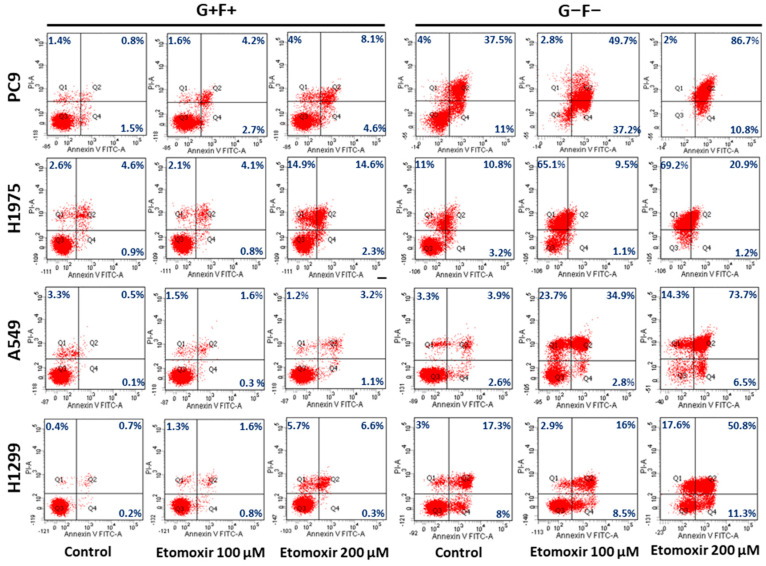
Lipids constitute an important energy source for lung cancer during starvation, and Etomoxir sensitizes lung cancer cells to starvation. Lung cancer cells are treated with Etomoxir either in normal growth medium (G+F+) or glucose- and serum-free medium (G−F−) for 48 h. Cell death was measured by Annexin V/PI staining using flow cytometry.

## Data Availability

Not applicable.

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
