# Peer review of "Lipid Droplets in Lung Cancers Are Crucial for the Cell Growth and Starvation Survival"

_ijms, 2022, doi:10.3390/ijms232012533_

Round 1

Reviewer 1 Report

The original research article is potentially interesting. However, I found several problems that preclude publishing in the present form. More detailed explanations and additional experiments are requested.

1.     In the present study, the authors used lung adenocarcinoma cell lines as tumor cells. On the other hand, the authors used normal lung fibroblast cell, WI-38, as control cell for comparison. However, WI-38 cell is not epithelial cell. Therefore, WI-38 cell is not suitable as control for adenocarcinoma cells.

2.     In line 110-112, the authors described that “the expression of low-density lipoprotein receptor (LDLR), low-density lipoprotein receptor-related proteins 5 and 6 (LRP5 and LRP6) are also upregulated in several lung cancer cells compared with the normal lung fibroblast cell, WI-38”. However, the expression of LDLR is downregulated in several lung cancer cells compared with WI-38 cell.

3.     The authors should identify the lipid accumulated in lung adenocarcinoma in this study. Are the accumulated lipids triglycerides, cholesterol, or phospholipids?

4.     In the Fig. 3, the authors showed the accumulation of triglycerides in lung adenocarcinoma cells. However, the authors showed that lipid accumulation in lung adenocarcinoma by LDL stimulation in Fig. 1. In general, it is thought that cholesterol accumulation is seen with LDL stimulation. The enhancing effect on triglyceride accumulation by LDL stimulation is observed in lung adenocarcinoma? LDL stimulation also promote the accumulation of cholesterol?

5.     The authors need to prove whether knockdown of PLIN3 reduces LD formation in adenocarcinoma cells.

6.     In the Figure 4, lung adenocarcinoma cells are induced cell proliferation under lipoprotein-deficient medium culture condition? If lung adenocarcinoma cells are induced cell proliferation under LDL-free condition, what is the source of cholesterol, a component of the cell membrane?

7.     In the Figure 5, DGAT1 and DGAT2 inhibitors induce cell death? The authors should examine the effect of DGAT1 and DGAT2 inhibitors on apoptosis and necrosis in lung adenocarcinoma cells. 

8. In line 226-229, the authors described that “As shown in the left half of Figure 6, the sensitivities of the examined cells to Etomoxir were quite mild under a normal growth medium, but massive cell deaths were induced by Etomoxir treatment under starvation, especially the low LD-containing cell, PC9 (top right side of Figure 6).” 

Author Response

Responses to the comments from reviewer 1

Dear reviewer:

Thank you for giving us these valuable comments and guiding us to further improve our manuscript. Our replies to these valuable questions and revisions for the manuscript are listed in the following. All revisions in the manuscript were marked with tracking changes. Hope these improvements could fully clarify the concerns of reviewers and receive final acceptance for publication.

  1. In the present study, the authors used lung adenocarcinoma cell lines as tumor cells. On the other hand, the authors used normal lung fibroblast cell, WI-38, as control cell for comparison. However, WI-38 cell is not epithelial cell. Therefore, WI-38 cell is not suitable as control for adenocarcinoma cells.

Reply:  Thanks for the reviewer’s valuable comment. Noncancerous cells, including stromal fibroblast, adjacent to the tumor part in patient samples are frequently adopted as normal parts in the pathology and immunohistology interpretation. We admit that a normal lung cell line of non-epithelial origin is not the best control cell line for the study.  With no normal lung epithelial cell line in hand and the easy propagation property of WI-38 cells,  the cell line was chosen as the control cell line in the current study.

  1. In line 110-112, the authors described that “the expression of low-density lipoprotein receptor (LDLR), low-density lipoprotein receptor-related proteins 5 and 6 (LRP5 and LRP6) are also upregulated in several lung cancer cells compared with the normal lung fibroblast cell, WI-38”. However, the expression of LDLR is downregulated in several lung cancer cells compared with WI-38 cell.

Reply: Thanks for the reviewer’s reminder. The expressions of LDLR in most of the examined lung cancer cells are lower than that of the normal lung cell line, WI-38. The statement is revised to “the expression of low-density lipoprotein receptor-related proteins 5 and 6 (LRP5 and LRP6) are also upregulated in several lung cancer cells compared with the control normal lung cell, WI-38”.

  1. The authors should identify the lipid accumulated in lung adenocarcinoma in this study. Are the accumulated lipids triglycerides, cholesterol, or phospholipids?

Reply: We do intend to find a reliable service provider to assay the lipid species in lipid droplets for these lung cancer cell lines, but the asking price is far from affordable to our budget.  Since the majority of lipid droplets in the examined lung cancer cell lines could be eliminated by DGAT1/2 inhibitors, triglycerides would likely be the most abundant lipid species within the lipid droplets of these lung cancer cells.

  1. In the Fig. 3, the authors showed the accumulation of triglycerides in lung adenocarcinoma cells. However, the authors showed that lipid accumulation in lung adenocarcinoma by LDL stimulation in Fig. 1. In general, it is thought that cholesterol accumulation is seen with LDL stimulation. The enhancing effect on triglyceride accumulation by LDL stimulation is observed in lung adenocarcinoma? LDL stimulation also promote the accumulation of cholesterol?

Reply: Thanks for the reviewer’s valuable comment. We admitted that it would be better to adopt a fluorescence-labeled VLDL to show the lipid uptake in these lung cancer cells. But we could find only fluorescence-labeled LDL at that time. Since apolipoprotein E (ApoE)  receptors, including VLDLR, LDL, and LRP, are all involved in the clearance of triglyceride in plasma. VLDLR deficiency in mice minimally affects plasma lipoproteins, and only results in a delay in the clearance of postprandial TG. These results indicated that LDLR could also involve in plasma TG clearance. For this reason, the fluorescence-labeled LDL was used to monitor the lipid uptake in lung cancer cell lines. The related information and references were added to the manuscript for clarity.

Reference:

  1. Triglyceride-rich lipoprotein metabolism in unique VLDL receptor, LDL receptor,

and LRP triple-deficient mice. J. Lipid Res. 2005. 46: 1097–1102.

  1. Chapter 72 - Liver Physiology and Energy Metabolism, Editor(s): Mark Feldman, Lawrence S. Friedman, Lawrence J. Brandt,Sleisenger and Fordtran's Gastrointestinal and Liver Disease (Ninth Edition),W.B. Saunders,2010,Pages 1207-1225.e3,

  1. The authors need to prove whether knockdown of PLIN3 reduces LD formation in adenocarcinoma cells.

Reply: Thanks for the reviewer’s valuable comment. PLIN2 and PLIN3 share a high level of sequence identity and are both ubiquitously expressed in different tissues and cells.  Knocking down PLIN3 alone could decrease but not eliminate the lipids droplets, probably due to the presence of PLIN2 (Hussain et. al. 2021).  To efficient elimination of lipid droplets, we adopted pharmacological depletion of lipid droplets using DGAT1/2 treatment.

Reference:

Hussain SS, Tran TM, Ware TB, Luse MA, Prevost CT, Ferguson AN, Kashatus JA, Hsu KL, Kashatus DF. RalA and PLD1 promote lipid droplet growth in response to nutrient withdrawal. Cell Rep. 2021;36(4):109451.

  1. In the Figure 4, lung adenocarcinoma cells are induced cell proliferation under lipoprotein-deficient medium culture condition? If lung adenocarcinoma cells are induced cell proliferation under LDL-free condition, what is the source of cholesterol, a component of the cell membrane?

Reply: The cell growth and proliferation of these lung cancer cell lines were significantly slowed down when cultured with a lipoprotein-depleted growth medium. The cell images in Figure 4 were shot 7 days after being cultured with a lipoprotein-depleted medium to reach a cell density comparable to cells cultured with a normal growth medium after 3 days. Despite a shortage of extracellular supply, the endogenous de novo lipid biosynthesis pathway should be able to support the lipid needs, including fatty acid and cholesterol, for cell proliferation under the lipoprotein-depleted condition.

  1. In the Figure 5, DGAT1 and DGAT2 inhibitors induce cell death? The authors should examine the effect of DGAT1 and DGAT2 inhibitors on apoptosis and necrosis in lung adenocarcinoma cells. 

Reply: DGAT1 and DGAT2  inhibitor treatment do not induce cell death significantly in these lung cancer cell lines since no roundup cells and cell debris were observed during the DGAT1/2 treatment. We think that the treatment only slows down the cell growth and proliferation of these lung cancer cell lines. 

  1. In line 226-229, the authors described that “As shown in the left half of Figure 6, the sensitivities of the examined cells to Etomoxir were quite mild under a normal growth medium, but massive cell deaths were induced by Etomoxir treatment under starvation, especially the low LD-containing cell, PC9 (top right side of Figure 6).” 

Reply: To make the readers more easily understand the result, the paragraph is modified to “As shown in Figure 6, the sensitivities of the examined cells to Etomoxir were quite mild under a normal growth medium (the second and third columns of Figure 6), but massive cell deaths were induced by Etomoxir treatment under starvation (the fifth and sixth columns of Figure 6).

Reviewer 2 Report

The manuscript reports a potentially important and interesting data. However, the paper has severe flaws and notable weaknesses. Moreover most of presented data are far from originality. Therefore, I cannot recommended publishing this article in the present form.

  Specific comments:   Major   1.Numerous studies have already demonstrated the role of altered: a) fatty acid biosynthesis, b) fatty acids transport and storage of lipids and c) fatty acids oxidation in lung cancer. For instance see recently published  review by Claudia Fumarola  et al in: Cells 2022, 413 or in review published in Critical Reviewers in Oncology/Hematology 2017, 112, 31-40. Thus, Authors should emphasize the most important (original) problems presented in this manuscript, which allow to accept this work in IJMS.      2. The conclusion is based only on study of level of protein proteins involve in: a) fatty acid biosynthesis, b) fatty acids transport and storage of lipid and c) fatty acids oxidation in: 9 cancer cells lines and 1 cell  line of normal human cell – fibroblast derived from lung (WI-38). This line presumably serve as a control. However, it is not clear, reading the manuscript, whether WI-38 serves as a control for studied cancer cells (for instance see legends to Figures 1 and 2, where WI-38 are treated as cancer cells). The authors should clarify this problem. Moreover, the questions arises why authors studied/selected: A 549, H292, H441. H460, H 1299, H1650, H 1975, HCC827, PC9 cancer cells lines to resolve the problem of lipid metabolism in lung cancer. What were the criteria for selection such cells lines.    3. Usually SREBP-1 gene expression is well correlate with ACLY, ACC and FASN genes expression. In Figs 1 and 2, I cannot see such correlation. Could you explain this discrepancy?   4. I wonder why authors measured only protein levels of lipogenic enzymes, but not mRNA and/or enzyme activity.   Minor: 1.Title should be slightly change (“normal” should be omitted): Proposed version: Lipid droplets in lung cancers are crucial for the cell growth and starvation survival.   2. ATP citrate lyase should be ACLY, but not ACYL ; page 2, line 54   3.Should be fatty acid synthase, but not fatty acid synthetase; page 3, lines100/101.   4. In phrase: Despite the pivotal role of fatty acids … or Lecitin –cholesterol acyltransferase (LCAT).( page 2, lines 63-69) … or Lecitin – cholesterol acyltransferase (LCAT) should be omitted, because free fatty acids are not involve in reaction catalyzed by LCAT.

Author Response

Responses to the comments from reviewer 2

Dear reviewer:

Thank you for giving us these valuable comments and guiding us to further improve our manuscript. Our replies to these valuable questions and revisions for the manuscript are listed in the following. All revisions in the manuscript were marked with tracking changes. Hope these improvements could fully clarify the concerns of reviewers and receive final acceptance for publication.

1.Numerous studies have already demonstrated the role of altered: a) fatty acid biosynthesis, b) fatty acids transport and storage of lipids and c) fatty acids oxidation in lung cancer. For instance see recently published  review by Claudia Fumarola  et al in: Cells 2022, 413 or in review published in Critical Reviewers in Oncology/Hematology 2017, 112, 31-40. Thus, Authors should emphasize the most important (original) problems presented in this manuscript, which allow to accept this work in IJMS.     

Reply: Thanks for the reviewer’s valuable comment. Despite the reprogramming of lipid metabolism and uptake system to meet the demand for the rapid cell growth and proliferation in lung cancer cells is well investigated, whether different lung cancer cells adopt identical mechanisms to ensure sufficient lipid supply and whether the lipid demand and supply match each other remain unclear. Results showed that despite frequent upregulation in de novo lipogenesis and lipid transporter system, different lung cancer cells adopt different proteins to acquire sufficient lipids, and the lipid supply frequently exceeds the demand as significant amounts of lipids stored in the lipid droplets could be found within lung cancer cells. Pharmacological depletion of LDs and inhibition of lipid utilization have great impacts on the cell growth and proliferation, and survival of lung cancer cells, respectively. These findings suggest that manipulation of lipid droplet formation or TAG storage in lung cancer cells could potentially decrease the progression of lung cancer, and would be interesting to the scientific community and readers of IJMS.  The abstract and the fourth paragraph of the introduction section are revised to emphasize these issues.

  1. The conclusion is based only on study of level of protein proteins involve in: a) fatty acid biosynthesis, b) fatty acids transport and storage of lipid and c) fatty acids oxidation in: 9 cancer cells lines and 1 cell  line of normal human cell – fibroblast derived from lung (WI-38). This line presumably serve as a control. However, it is not clear, reading the manuscript, whether WI-38 serves as a control for studied cancer cells (for instance see legends to Figures 1 and 2, where WI-38 are treated as cancer cells). The authors should clarify this problem. Moreover, the questions arises why authors studied/selected: A 549, H292, H441. H460, H 1299, H1650, H 1975, HCC827, PC9 cancer cells lines to resolve the problem of lipid metabolism in lung cancer. What were the criteria for selection such cells lines.   

Reply: Thanks for the reviewer’s valuable comment. In the current study, the expression of target proteins in the normal lung cell line, WI-38, was chosen as a reference to measure the expression changes of target proteins in other lung cancer cell lines. To clarify the issue, the figure legends of Figure 1 and Figure 2 and Th Results part were revised. Cell lines used in the study were chosen based on the prevalence rate of different oncogenic mutations in lung cancer. Since EGFR and KRAS driver mutations account for the most common driver mutation type in the east Asian population and Caucasian population, respectively, 4 EGFR mutation and 4 KRAS mutation lung cancer cell lines were chosen in the study. The mucin-producing low-grade mucoepidermoid carcinomas cell lines, H292, was also included to monitor the expressions of target proeins in cells with malignancy in-between the normal and malignant lung cell lines.

  1. Usually SREBP-1 gene expression is well correlate with ACLY, ACC and FASN genes expression. In Figs 1 and 2, I cannot see such correlation. Could you explain this discrepancy?  

Reply: Thanks for the reviewer’s valuable comment. The relative abundance of a protein would be determined by multiple mechanisms, including transcription, translation, and the stability of mRNA and protein. The mechanisms that account for the discrepancy between the expression levels of ACLY, ACC, and FASN at a similar upregulated expression level of SREBP-1 in different cell lines are not clear at this moment, but should be related to the expression levels of other transcription factors, or the stability of mRNA and protein. For example, it has been found that EGFR2 could upregulate ACC and FASN but not ACLY translationally (Yoon et.al. 2007), and different ubiquitin ligases preferentially degrade one or two of the three enzymes (Zhu et. al. 2022).

References

  1. Yoon S, Lee MY, Park SW, Moon JS, Koh YK, Ahn YH, Park BW, Kim KS. Up-regulation of acetyl-CoA carboxylase alpha and fatty acid synthase by human epidermal growth factor receptor 2 at the translational level in breast cancer cells. J Biol Chem. 2007, 282(36):26122-31.
  2. Zhu Y, Lin X, Zhou X, Prochownik EV, Wang F, Li Y. Posttranslational control of lipogenesis in the tumor microenvironment. J Hematol Oncol. 2022.15(1):120.

  1. I wonder why authors measured only protein levels of lipogenic enzymes, but not mRNA and/or enzyme activity.  

Reply: Thanks for the reviewer’s valuable comment. Despite the enzymatic measurement would be the best strategy to monitor the real activity of an enzyme in biological samples, the measurement could be very difficult to perform if there is no simple strategy to monitor the change of substrate or product, or require special instruments.  On the other hand,  measuring the abundance of protein by monitoring the level of the translation template of enzyme, mRNA, can often be affected by translational control and mRNA stability issues. These could make the discrepancy between the expression levels of mRNA and protein frequently observed and measuring protein abundance by specific antibodies widely adopted in biomedical research.

5a.Title should be slightly change (“normal” should be omitted): Proposed version: Lipid droplets in lung cancers are crucial for the cell growth and starvation survival.  

Reply: Thanks for the reviewer’s valuable suggestion. The title is revised.

5b. ATP citrate lyase should be ACLY, but not ACYL ; page 2, line 54  

Reply: We apologize for the negligence and thanks for the reminders. The symbol for ATP citrate lyase is carefully scrutinized in the manuscript and revised to ACLY.

5c.Should be fatty acid synthase, but not fatty acid synthetase; page 3, lines100/101.

Reply: Thanks for the reviewer’s kind reminders. A synthetase utilized nucleoside triphosphate during reaction but a synthase does not.  Fatty acid synthetase is revised to fatty acid synthase.

5d. In phrase: Despite the pivotal role of fatty acids … or Lecitin –cholesterol acyltransferase (LCAT).( page 2, lines 63-69) … or Lecitin – cholesterol acyltransferase (LCAT) should be omitted, because free fatty acids are not involve in reaction catalyzed by LCAT.

Reply: Thanks for the reviewer’s valuable comment. The sentences were deleted.

Round 2

Reviewer 1 Report

Thank you for your responses to my previous comments. I understood your responses. The content of your responses to my previous comments (Comment 1, 4, 5) needs to be described in discussion section in the manuscript.

Author Response

Dear reviewer:

Thank you for giving us these valuable comments and guiding us to further improve our manuscript. We admit that there are some insufficiencies in the study and are glad our responses and revisions could clarify your concerns. The contents of the responses to your previous comments (Comments 1, 4, 5) were added to the second last paragraph of the discussion section to disclose the limitation and insufficiency of the study. We promise that we will keep on investigating the roles of lipid droplets in cancer and find out strategies to fight cancer metabolically.

Reviewer 2 Report

The manuscript has been significantly improved. I have no further comments.

Author Response

Dear reviewer:

Thank you for giving us these valuable comments and guiding us to further improve our manuscript. We admit that there are some insufficiencies in the study and are glad our responses and revisions could clarify your concerns. We promise that we will keep on investigating the roles of lipid droplets in cancer and find out strategies to fight cancer metabolically.